

# The intensification of metallic layered phenomena above thunderstorms through the modulation of atmospheric tides

Bingkun Yu[1,2], Xianghui Xue[1,2,3], Chengling Kuo[4], Gaopeng Lu[5,6], Xiankang Dou[1,2], Qi Gao[1,2], Jianfei Wu[1,2], Mingjiao Jia[1,2], Chao Yu[1,2], and Xiushu Qie[5,6]

[1]CAS Key Laboratory of Geospace Environment, Department of Geophysics and Planetary Sciences, University of Science and Technology of China, Hefei, China
[2]Mengcheng National Geophysical Observatory, School of Earth and Space Sciences, University of Science and Technology of China, Hefei, China
[3]Synergetic Innovation Center of Quantum Information and Quantum Physics, University of Science and Technology of China, Hefei, China
[4]Institute of Space Science, National Central University, Jhongli, Taiwan
[5]Key Laboratory of Middle Atmosphere and Global Environment Observation, Institute of Atmospheric Physics, Chinese Academy of Sciences, Beijing, China
[6]Collaborative Innovation Center on Forecast and Evaluation of Meteorological Disasters, Nanjing University of Information Science and Technology, Nanjing, China

**Correspondence:** Xianghui Xue (xuexh@ustc.edu.cn)

**Abstract.** We present a multi-instrument experiment to study the effects of tropospheric thunderstorms on the mesopause region and the lower ionosphere. The sodium (Na) lidar observation and the ionospheric observation by two digital ionospheric sounders are used to study the variation of the neutral metal atoms and metallic ions above thunderstorms. The enhanced ionospheric sporadic $E$ layer with a downward tidal phase is observed followed by a subsequent intensification of neutral Na density with an increase of 600 cm$^{-3}$ in the mesosphere. In addition, the Na neutral chemistry and ion-molecule chemistry reactions are considered in the Na chemistry model to simulate the dynamical and chemical coupling processes in the mesosphere and ionosphere above thunderstorms. The enhanced Na layer in the simulation using the ionospheric observation as input is in agreement with the Na lidar observation. We find that the intensification of metallic layered phenomena above thunderstorms is associated with the atmospheric tides, as a result of the troposphere-mesosphere-ionosphere coupling.

## 1 Introduction

The ionospheric sporadic $E$ ($E_s$) layer and neutral sodium (Na) layer are both metallic layered phenomena of great interest in the Mesosphere/Lower Thermosphere (MLT) region between neutral and ionized atmosphere. The $E_s$ layers are thin and dense patches of metallic ions at altitudes between 90 and 130 km. The metallic ions are from the meteoric ablation (Von Zahn et al., 2002; Plane, 2003; Carrillo-Sánchez et al., 2015; Plane et al., 2015). The metallic $E_s$ formation relies on the vertical wind shear provided by the tidal wind, i.e., mostly the diurnal and semidiurnal tides (Whitehead, 1961, 1989; Mathews, 1998; Haldoupis, 2012). The $E_s$ descends with the vertical downward tidal phase until it weakens and then becomes depleted below 100 km. The neutralization of Na$^+$ occurs through three-body reactions needed for the metallic ions to cluster ions, followed by increased



dissociative electron recombination. The lifetime of $Na^+$ rapidly decreases from 8.7 h at 100 km to only a few minutes at 90 km (Plane et al., 2015). Therefore, the mesospheric sodium exists as layers of neutral atoms at 80–105 km altitudes with a peak density of $10^3$–$10^4$ $cm^{-3}$ near 92 km. There are also high temporal and spatial correlations between the $E_s$ and sporadic Na layer (Dou et al., 2010; Xue et al., 2013).

The coupling of tropospheric thunderstorms with the upper atmosphere and ionosphere has been known for many years since the pioneering work of Wilson (1924). A thunderstorm can disturb the MLT region through convective atmospheric gravity waves (GWs) (Sentman et al., 2003; Yue and Lyons, 2015) and lightning-induced transient electromagnetic phenomena (Pasko et al., 1997; Davis and Johnson, 2005; Lu, 2006; Cheng et al., 2007; Davis and Lo, 2008; Lu et al., 2011). The speculative connection between thunderstorms and $E_s$ layer was proposed in the 1930s (Watson-Watt, 1933; Ratcliffe and White, 1934)

and was first reported in Nature magazine based on a statistical superposed epoch analysis (SEA) from a very large dataset during the period from 1993 to 2003 (Davis and Johnson, 2005). It was the first paper to reveal an enhancement of $E_s$ layer which occurs ∼6 h and ∼30 h after lightning. After that, more studies applied the same methodology of SEA on lightning and $E_s$ layer measurements using the hourly lightning events as trigger times (Johnson and Davis, 2006; Davis and Lo, 2008; Barta et al., 2013; Yu et al., 2015; Barta et al., 2017). Nowadays, the mechanism responsible for the ionospheric $E_s$ perturbations

above thunderstorms is not well understood. A serious difficulty in explaining the lightning-induced coupling phenomena is the observed large time lag between the most statistically significant $E_s$ response and lightning trigger events. The time lag ranges from several hours to more than 30 hours (Davis and Johnson, 2005; Johnson and Davis, 2006; Yu et al., 2015). The recent work reported a novel observation of the meteoric metals in the mesosphere, atomic Na layer that is significantly intensified ∼19 h after lightning using SEA method (Yu et al., 2017). Nearly all these studies of lightning-induced effects on the metallic

layered phenomena were statistical results from the SEA method, and it is difficult to explain by the GWs or electrodynamic effects induced by lightning.

     As mentioned above, the enhancement of Na layer is closely related to the occurrence of $E_s$ layer. In our study, a multi-instrument experiment combined with the observation of neutral metal atoms and metallic ions is carried out to explore the lightning-$E_s$-Na relations. It is found that the lightning-induced enhancement of Na layer and $E_s$ layer are associated with

atmospheric tides, as a result of the troposphere-mesosphere-ionosphere coupling. Besides, the Na cluster ion chemistry is also considered in the process of the intensification of metallic layered phenomena. A Na chemistry model using the ionospheric $E_s$ observation as input is used to compare the Na density in simulation with the Na lidar observation. We further discuss the dynamical and chemical processes in the $E_s$ layer and Na layer during thunderstorms, suggesting the possible role of thunderstorms in this coupling of atmosphere and ionosphere.

## 30   2   Data and Observations

The World-Wide Lightning Location Network (WWLLN) is a global lightning detection system (Rodger et al., 2006). It has a relatively high detection efficiency in east Asia. At present, it is one of the best suited global lightning detection systems for investigating the location of the intense lightning (>50 kA) and its influence on the upper atmosphere.





The neutral metal Na atom layers are measured by a broadband dye-laser-based Na resonance fluorescence lidar at Haikou, China (20.0°N, 110.3°E), as part of the Chinese Meridian Project (Wang, 2010). The lidar has been routinely operated since 2010. A total of 1,577 h Na layer observations on 197 nights from 2010 to 2013 are available.

There are two digisondes near the Haikou Na lidar station, i.e., the Digital Portable Sounder 4D (DPS-4D) digisonde at Sanya
(18.3°N, 109.6°E) and the Royal Meteorological Institute digital ionosonde (Lowell DGS-256) at Fuke (19.4°N, 109.0°E). These ionospheric observations can provide the simultaneous information of metallic ions during the routine Na lidar operation.

It has been previously revealed that there is a relationship between neutral metal Na layer and thunderstorms, that is an enhancement of neutral Na layer in response to lightning (Yu et al., 2017). The lightning-induced statistically enhanced Na layer events distributed on 28 thunderstorm nights are further analyzed here to study the possible mechanism, on the basis of
the Na lidar data and ionospheric data from two digisondes. Figure 1a shows the residual of neutral metal Na median density for 150 hours either side of the lightning trigger events, after removing the diurnal trend (the difference of median response of Na density and median response of Gaussian random Na density after lightning trigger times in Yu et al. (2017)). The 99.9 % significant level is contoured as the solid white lines (i.e. the probability of the enhancement arose accidentally is 0.001). There are some other small patches of value exceeding the 99.9 % level because of signal-to-noise ratio at higher and lower heights.
However, the Na layer is obviously intensified up to $500\,\mathrm{cm}^{-3}$ after lightning. The zoom-in window between 0 and 25 h plotted in red is shown in Figure 1b. The enhancement of Na layer is evident at 17–24 h, 85–98 km after lightning with the maximum increase of $\sim$600 cm$^{-3}$ at t=19 h, at an altitude of 93 km.

There are high time and space correlations between the $E_s$ and sporadic Na layer (Dou et al., 2010). Figure 2a shows all the time series of hourly ionospheric $E_s$ layer and Na layer observations on 28 thunderstorm nights at Sanya. The time
ranges from 0 to 25 hours after lightning. The lower panel is the averaged WWLLN strokes per hour as a indicator of the intensity of thunderstorms. The average duration of thunderstorms is $\sim$13 h (from 10.11 Cts/hour at t=0 h to 15.43 Cts/hour at t=13 h), much longer than usual continental thunderstorms. There are more intense lightning strokes over the coastal Haikou lidar station on the Hainan island, with the 2,534.15 J mean stroke energy, 1,197.20 J median stroke energy, 31.3% high energy strokes ($>$2,000 J) and 13.2% low energy strokes ($<$400 J), compared with the continental area, e.g., Beijing with the 2,128.47 J
mean stroke energy, 632.38 J median stroke energy, 19.3% high energy and 34.4% low energy strokes. Although the statistical enhancement of Na density occurs at t=19 h, the observed time that Na layer reaches the maximum density differ each night. The occurrence of the 28 maximum Na densities was marked by red stars. The profiles of $E_s$ height are plotted in grey lines. The occurrence rate of $E_s$ is plotted in red dashed lines, and it can be found that the $E_s$ layer occurs more frequently during thunderstorms. The relative change of critical frequency, $f_oE_s$ ($(f-\overline{f})/\overline{f}$) is shown in blue dotted and solid lines, in which $\overline{f}$ is
the averaged background frequency of $f_oE_s$. To minimize the effects of the height on the frequency, the averaged background frequency is defined as the mean frequency of the dataset within $\pm$ 0.5 km of the $E_s$ height at each point. Blue solid lines are used when two consecutive points are positive, and blue dotted lines are used otherwise. In agreement with previous studies (Davis and Johnson, 2005), the $E_s$ layer is statistically enhanced. Both the occurrence rate and relative change of $E_s$ vary with the development of the underlying thunderstorm. The variation of $E_s$ is positively related to the averaged WWLLN strokes rate.
At t=0–8 h, with the increase of the lightning strokes occurrence rate, the $E_s$ layer becomes more dense and frequent. The peak





of the observed relative $f_oE_s$ reaches $\sim$40 % ($f_oE_s$=5.03 MHz) and the maximum of occurrence rate is $\sim$0.8. Between t=8 and 13 h, both the critical frequency and occurrence rate of $E_s$ layer decrease along with the decrease of thunderstorm activity. Besides, the height of $E_s$ layer decreases until it descends below 100 km before the Na density reaches the maximum value. The $E_s$ height descent is mainly driven by tidal winds which provide the vertical wind shear to shepherd the ions downwards

through their tidal phase velocity propagation. To clearly show the tendency, the average height of $E_s$ is plotted in a green solid line, and it is found that the height of $E_s$ shows a diurnal variation, mainly controlled by the diurnal tide.

Figure 2b shows data from another digisonde at Fuke near the Na lidar. It shows the similar tendency of tide-period height decrease, a higher occurrence rate of $E_s$ and increased relative $f_oE_s$ before the maximum Na densities occur during thunderstorms. Compared with Figure 2a, the occurrence rate of $E_s$ in Figure 2b is relatively low. It is because the DGS-256 digisonde

ionograms from Fuke were automatically scaled, while the DPS-4D digisonde ionograms from Sanya were manually scaled to avoid unpredictable poor automatic scaling and identify untypical $E_s$ layer from disturbed ionograms. Furthermore, Figure 2b shows the $E_s$ height is controlled not only by the diurnal tide but also by the semidiurnal tide.

## 3    Chemical Simulation

In addition to the dynamics in the $E_s$ and Na layers which are mostly controlled by the diurnal and semidiurnal tides, the

role of chemistry in the MLT region should also be considered in the decay of descending $E_s$ layer and related occurrence of intensified Na layer. To study the Na reactions with the $E_s$ layer, a Na chemistry model is used to simulate the dynamical and the chemical processes of the enhanced Na layer through the modulation of atmospheric tides. The neutral and ionic gas-phase chemistry schemes in our Na model are taken from the reactions of Na neutral chemistry and ion-molecule chemistry with their rate coefficients in a recent atmospheric chemistry review (Plane et al., 2015), with more details shown in the appendix

table. In the model, the main long-lived Na species, Na, Na$^+$, NaHCO$_3$ and e$^-$ are calculated from the solution of continuity equations below, while other short-lived intermediates are considered as steady-state concentration (Plane, 2004). The time tendencies of Na, NaHCO$_3$, Na$^+$ and e$^-$ are described as follows:

$$\frac{d[Na]}{dt} = I_{abl} + A[NaHCO_3] + B[Na^+] - (C+D)[Na] - \nabla\Phi^{Na} \tag{1}$$

$$\frac{d[NaHCO_3]}{dt} = D[Na] - A[NaHCO_3] - 2k_{12}[NaHCO_3]^2 - \nabla\Phi^{NaHCO_3} \tag{2}$$

$$\frac{d[Na^+]}{dt} = C[Na] - B[Na^+] - \nabla\Phi^{Na^+} \tag{3}$$


$$\frac{d[e^-]}{dt} = -B[Na^+], \tag{4}$$

where $I_{abl}$ is the Na injection rate according to height profiles of the Na input rate from meteor ablation, calculated for the Long Duration Exposure Facility meteoroid size distribution (McBride et al., 1999) (85 % reduce in all mass ranges with a





global input of 12.1 t d$^{-1}$) with the peak at an altitude of 94 km (Plane, 2004).

$$A = k_9[H],$$

$$B = k_{22}[N_2][M]\left(\frac{k_{24}([CO_2]+[H_2O])+k_{29}[e^-]+\frac{k_{29}[e^-]k_{25}[O]}{k_{26}[O]+k_{27}[N_2]+k_{28}[O_2]+k_{29}[e^-]}}{k_{24}([CO_2]+[H_2O])+k_{29}[e^-]+k_{25}[O]\frac{k_{26}[O]+k_{28}[O_2]+k_{29}[e^-]}{k_{26}[O]+k_{27}[N_2]+k_{28}[O_2]+k_{29}[e^-]}}\right)+$$

$$k_{23}[CO_2][M]+k_{30}[e^-],$$

$$C = k_{20}[O_2^+]+k_{21}[NO^+],$$

$$D = (k_1[O_3]+k_{10}[O_2][M])\left(\frac{k_4[H_2]+k_6[H_2O]}{k_2[O]+k_3[O_3]+k_4[H_2]+k_5[H_2]+k_6[H_2O]}\right)\left(\frac{k_8[CO_2][M]}{k_7[H]+k_8[CO_2][M]}\right).$$

We neglect the day-time photochemical reactions, to study the process of enhanced Na layer associated with the $E_s$ layer on thunderstorm nights. The Na model is one-dimensional, from 50–140 km with an altitude resolution of 2 km. Equations (1)–(4) are integrated with a 1 min time step. $\nabla\Phi^X$ is the divergence of the species X vertical flux (Plane, 2004). $\Phi^X$ is a function of height z:

$$\Phi^X = -K_{zz}\left(\frac{\partial[X]}{\partial z}+[X]\left(\frac{1}{H}+\frac{1}{T}\frac{\partial T}{\partial z}\right)\right), \tag{5}$$

where $K_{zz}$ is the vertical eddy diffusion coefficient, $4\times10^5$ cm$^2$s$^{-1}$. For the general nighttime condition over Haikou in July 2012, the neutral compositions, ions and temperature are obtained from the WACCM with the metal chemistry included (Feng et al., 2013).

The model runs over 30 days for all species to reach steady state, and then the ionospheric observations of digisonde at Fuke, near the Na lidar, from t=0 to 25 h are input to drive the Na model. The $E_s$ layer is assumed to be a Gaussian height profile ($\sigma$=3 km) with maximum electron density of $f_oE_s$. The ionospheric observations is input with 5 minutes time interval using linear interpolation between two adjacent observed values. Profiles of e$^-$ and Na$^+$ ($\sim$4% of metallic ions measured by in-situ rocket flights (Kopp, 1997)) are input according to the ionospheric $E_s$ observation. The variation of height of the $E_s$ layer is also considered as the dynamical process of plasma in the Na chemistry model. The term of $\boldsymbol{w}\cdot\nabla[Na]$ is added in the left of (1) as the vertical transport of Na atoms (Xu et al., 2000). The vertical wind for diurnal and semidiurnal tides is retrieved from the Global-Scale Wave Model 2009 (GSWM-09) tidal climatologies (Zhang et al., 2010a, b). The background of concentrations of chemical species and simulated density profiles of Na and Na$^+$ at t=0 h are shown in Figure 3a. Figures 3b–d are the simulation results from the Na model with the diurnal tide, diurnal+semidiurnal tides and 2$*$ (diurnal+semidiurnal) tides. Figure 3b gives the simulated Na density modulated by a diurnal tide. The main Na layer is around 2,400 cm$^{-3}$, which is consistent with the averaged Na lidar observations at Haikou. With a decrease in height and an increase in $f_oE_s$ of $E_s$ layer, the Na layer is obviously enhanced from 98 km to 88 km. The maximum is $\sim$2,800 cm$^{-3}$ at t=20 h, and at altitude of 94 km. The increase of Na density in the simulation is $\sim$400 cm$^{-3}$. Figure 3c is the simulated Na density in consideration of the vertical transport with diurnal and semidiurnal tides. Atmospheric tides play dominant roles in the dynamical process of the neutral metal Na layer. Weaker diurnal and semidiurnal perturbations in the Na layer structure or abundance could be found. Note that the condition of the Na chemistry model is driven by the ionospheric $E_s$ observation and the monthly GSWM-09





tidal climatologies. No tuning parameters were in the simulation. Only monthly averaged diurnal and semidiurnal tides from GSWM are considered in the Na chemistry model and other tidal components and GWs sources are neglected. The penetrative convection above thunderstorms can generate GWs that propagate vertically. Tidal backgrounds in the mesosphere could be reinforced through nonlinear interactions with the convectively generated GWs, and GWs could act to enhance the diurnal tide

amplitude above 90 km (Liu and Hagan, 1998; Liu et al., 2013). Therefore, the GSWM tides in the simulation may be relatively weaker as a result of an underestimate of tidal amplitude. Then an increased tidal amplitude is used in the simulation Figure 3d. The Na layer modulation of atmospheric tides is more evident with $2*$ (diurnal+semidiurnal) tides. In the downward motion of the lightning-induced enhanced $E_s$ layer with the tidal vertical phase, the Na layer intensifies while metallic $Na^+$ ions and electrons start weakening. The $Na^+$ ions are neutralized through three-body reactions, followed by a subsequent enhancement

of dissociative electron recombination. The Na layer density is enhanced at 18–20 h with a tidal downward phase and then transported upward at 20–25 h, with the modulation of the diurnal and semidiurnal tides.

The enhanced Na density in the simulation is consistent with the Na lidar observation in Figure 1b. This could prove the possible mechanism for the intensification of Na layer ∼19 h after lightning. The lightning-induced enhanced $E_s$ layer descends with the tidal phase below 100 km. Then the $E_s$ depletes via enhanced three-body collisional recombination to form

the enhanced metal Na layer above thunderstorms. The intensification of $E_s$ layer and Na layer are associated metallic layered phenomena through the modulation of atmospheric tides during thunderstorms. The modulation of atmospheric tides dominate the variation of the $E_s$ layer. As mentioned above, the statistical results of various studies using the SEA method have exhibited lightning discharges effect on $E_s$ layers. The lightning-induced response of $E_s$ layers is found to occur several to many hours after lightning, which makes it difficult to explain these phenomena by thunderstorms generated GWs or lightning-emitted

electromagnetic pulses. In fact, the 24-h diurnal tidal modulation can be found in the very large time lags observed between the response of $E_s$ layer and lighting, e.g., $E_s$ intensifications were observed to occur ∼6 h and ∼30 h after lightning (Davis and Johnson, 2005). In this paper, Figure 2 shows the similar lag time that the peak of relative $f_oE_s$ occurs ∼8 h after lightning, comparable to ∼6 h. The atmospheric tide is a dominant dynamical process in the MLT and plays dominant roles in the formation and dynamical processes of the intensification of metallic layered phenomena above thunderstorms.

**4   Discussion and Conclusions**

A thunderstorms influences the upper atmosphere and the ionosphere to make the ionospheric ionization (Pasko et al., 1997; Inan et al., 1991; Davis and Johnson, 2005; Davis and Lo, 2008; Yu et al., 2015) and the ionospheric depletions (Barta et al., 2017). Davis and Johnson (2005) first identified the influence of thunderstorms on the ionospheric $E_s$ layer around 100 km using the SEA method and hypothesized that the metal atoms may be related to thunderstorms. It remains possible that the

observed lightning-induced ionization enhancement in the $E_s$ is associated with the TLEs (Johnson and Davis, 2006). The magnitude of the perturbation to the $E_s$ layer is also associated with the intensity of lightning (Yu et al., 2015). The electrical processes associated with lightning and sprite above thunderstorms may have roles in the coupling of the atmosphere and ionosphere (Rycroft, 2006) .



Tropospheric thunderstorms can also affect the upper atmosphere through GWs induced in the lower atmosphere (Sentman et al., 2003). Atmospheric tides modulate the dynamical process in the MLT and play important roles in the dynamical process of the metallic layered phenomena, i.e., the $E_s$ layer (Whitehead, 1961, 1989; Haldoupis, 2012) and the neutral metal Na layer (She et al., 2002; Dou et al., 2010, 2013; Liu et al., 2013). In principle, GWs could also alter and influence the tidal forcing (Haldoupis, 2012). Tidal backgrounds in the mesosphere could be reinforced through nonlinear interactions with GWs (Liu and Hagan, 1998; Haldoupis et al., 2004). Besides, the elve luminosity with pronounced stripes was also found to be modulated by the thunderstorm-induced GWs (Yue and Lyons, 2015). Therefore, the thunderstorms generated GWs would be expected to act to influence the coupling processes of atmosphere and ionosphere. In Figure 4, a schematic diagram is shown to illustrate the proposed mechanism for the thunderstorm influence on the ionospheric $E_s$ layer and neutral metal Na layer.

In this study, we present a combination of the observational and numerical modelling results. The results presented here show robust evidence that the thunderstorm electrical effects accelerate and reinforce this process from metallic $Na^+$ ions to neutral Na atoms. The lightning-induced intensification of ionospheric $E_s$ layer is followed by the occurrence of the enhancement of neutral metal Na layer above thunderstorms. We conclude that the increase in the concentration of neutral Na atoms 19 h after lightning could be attributed to the enhanced ionospheric $E_s$ layer during thunderstorms. Atmospheric tides control the dynamical process in the MLT region and the $E_s$ layer descends with a diurnal tidal downward phase. The descending lightning-induced enhanced $E_s$ layer depletes below 100 km. In this downward motion, the three-body collisions become more effective and the chemical lifetime of $Na^+$ decreases. It efficiently enhances the dissociative electron recombination, to form the enhanced neutral Na layer above thunderstorms.

However, the whole coupling processes of atmosphere and ionosphere above thunderstorms has not been comprehensively proven, and it is left as an open question for what is the influence of thunderstorms on the upper atmosphere and what is the possible connection between the TLEs and the response of metallic layered phenomena to thunderstorms. More investigations are needed to further study these questions in observations and modelling.



**Table A1.** Reactions of Sodium Neutral Chemistry and Ion-molecule Chemistry with their Rate Coefficients in the Model (Plane et al., 2015)

| | Reaction | Rate coefficient |
|---|---|---|
| | **Neutral Chemistry** | |
| R1 | $Na + O_3 \rightarrow NaO + O_2$ | $1.1 \times 10^{-9} exp(-116/T)$ |
| R2 | $NaO + O \rightarrow Na + O_2$ | $(2.2 \times 10^{-10})(T/200)^{1/2}$ |
| R3 | $NaO + O_3 \rightarrow Na + 2O_2$ | $3.2 \times 10^{-10} exp(-550/T)$ |
| R4 | $NaO + H_2 \rightarrow NaOH + H$ | $1.1 \times 10^{-9} exp(-1100/T)$ |
| R5 | $NaO + H_2 \rightarrow Na + H_2O$ | $1.1 \times 10^{-9} exp(-1400/T)$ |
| R6 | $NaO + H_2O \rightarrow NaOH + OH$ | $4.4 \times 10^{-10} exp(-507/T)$ |
| R7 | $NaOH + H \rightarrow Na + H_2O$ | $4 \times 10^{-11} exp(-550/T)$ |
| R8 | $NaOH + CO_2 (+M) \rightarrow NaHCO_3$ | $(1.9 \times 10^{-28})(T/200)^{-1}$ |
| R9 | $NaHCO_3 + H \rightarrow Na + H_2CO_3$ | $(1.84 \times 10^{-13})T^{0.777} exp(-1041/T)$ |
| R10 | $Na + O_2 (+M) \rightarrow NaO_2$ | $(5.0 \times 10^{-30})(T/200)^{-1.22}$ |
| R11 | $NaO_2 + O \rightarrow NaO + O_2$ | $5.0 \times 10^{-10} exp(-940/T)$ |
| R12 | $2NaHCO_3 (+M) \rightarrow (NaHCO_3)_2$ | $(8.8 \times 10^{-10})(T/200)^{-0.23}$ |
| | **Ion-Molecule Chemistry** | |
| R20 | $Na + O_2^+ \rightarrow Na^+ + O_2$ | $2.7 \times 10^{-9}$ |
| R21 | $Na + NO^+ \rightarrow Na^+ + NO$ | $8.0 \times 10^{-10}$ |
| R22 | $Na^+ + N_2 (+M) \rightarrow Na \cdot N_2^+$ | $(4.8 \times 10^{-30})(T/200)^{-2.2}$ |
| R23 | $Na^+ + CO_2 (+M) \rightarrow Na \cdot CO_2^+$ | $(3.7 \times 10^{-29})(T/200)^{-2.9}$ |
| R24 | $Na \cdot N_2^+ + X \rightarrow Na \cdot X^+ + N_2 (X = CO_2, H_2O)$ | $6 \times 10^{-10}$ |
| R25 | $Na \cdot N_2^+ + O \rightarrow Na \cdot O^+ + N_2$ | $4 \times 10^{-10}$ |
| R26 | $NaO^+ + O \rightarrow Na^+ + O_2$ | $1 \times 10^{-11}$ |
| R27 | $Na \cdot O^+ + N_2 \rightarrow Na \cdot N_2^+ + O$ | $1 \times 10^{-12}$ |
| R28 | $Na \cdot O^+ + O_2 \rightarrow Na^+ + O_3$ | $5 \times 10^{-12}$ |
| R29 | $Na \cdot Y^+ + e^- \rightarrow Na + Y (Y = N_2, CO_2, H_2O, O)$ | $(1 \times 10^{-6})(T/200)^{-1/2}$ |
| R30 | $Na^+ + e^- \rightarrow Na + h\nu$ | $(3.9 \times 10^{-12})(T/200)^{-0.74}$ |



*Author contributions.* BY and XX designed the research, performed data analysis and wrote the manuscript. CK and GL contributed significantly to the comments on an early version and improvement in the manuscript. XD and QG derived and processed Na density data from raw photon count profiles. GL and XQ provided WWLLN lightning data. JW, MJ and CY contributed to discussion of the results and preparation of the manuscript. All authors discussed the results and commented on the manuscript at all stage.

5   *Competing interests.* The authors declare that they have no conflict of interest.

*Acknowledgements.* We acknowledge the data used in this paper from the Chinese Meridian Project, the Solar-Terrestrial Environment Research Network (STERN), and the World-Wide Lightning Location Network (WWLLN). We also acknowledge the ionospheric data at Sanya provided by Prof. Baiqi Ning and Dr. Lianhuan Hu in Institute of Geology and Geophysics, Chinese Academy of Sciences, the Whole Atmosphere Community Climate Model (WACCM) data over Haikou provided by Dr. Wuhu Feng in University of Leeds,

10  Leeds, UK and the Global-Scale Wave Model 2009 (GSWM-09) data provided by Dr. Xiaoli Zhang in University of Colorado, Boulder, Colorado, USA. This work is supported by the National Natural Science Foundation of China (41774158, 41474129, 41421063, 41574179,41704149,41704148,41804147), the open research project of CAS Large Research Infrastructures, the Youth Innovation Promotion Association of the Chinese Academy of Sciences (2011324), the Ministry of Science and Technology grant in Taiwan (MOST 106-2119-M-008-012), "The Hundred Talents Program" of Chinese Academy of Sciences (2013068) and the Fundamental Research Fund

15  for the Central Universities.

The ionospheric data at Sanya are available from Data Center for Geophysics, Data Sharing Infrastructure of Earth System Science, National Science & Technology Infrastructure of China (geospace.geodata.cn). The Na lidar data at Haikou and the ionospheric data at Fuke are available from Data Centre for Meridian Space Weather Monitoring Project (data.meridianproject.ac.cn). The WWLLN lightning data are available from World Wide Lightning Location Network (wwlln.net).



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



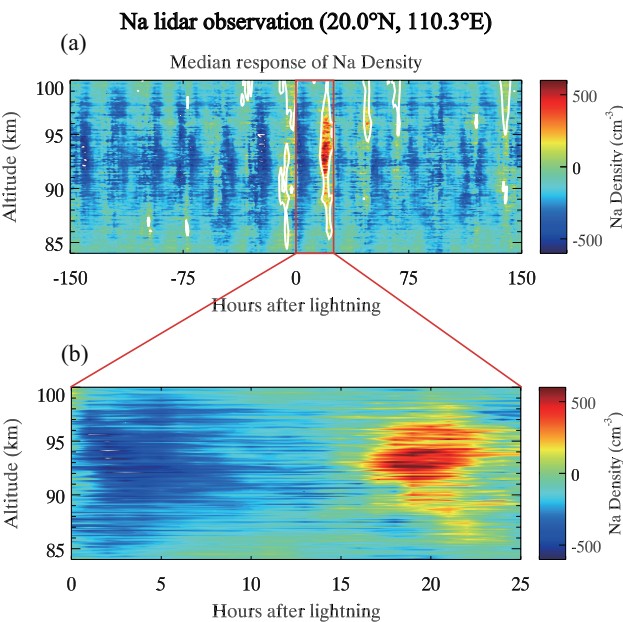

**Figure 1.** The enhanced neutral metal Na layer in response to lightning observed by the Na lidar at Haikou (20.0°N, 110.3°E). (a) The residual of Na median density from superposed epoch analyses 150 h before and after lightning, after removing the diurnal trend. (b) The zoom-in window in (a): the variation of Na density between 0 and 25 h.

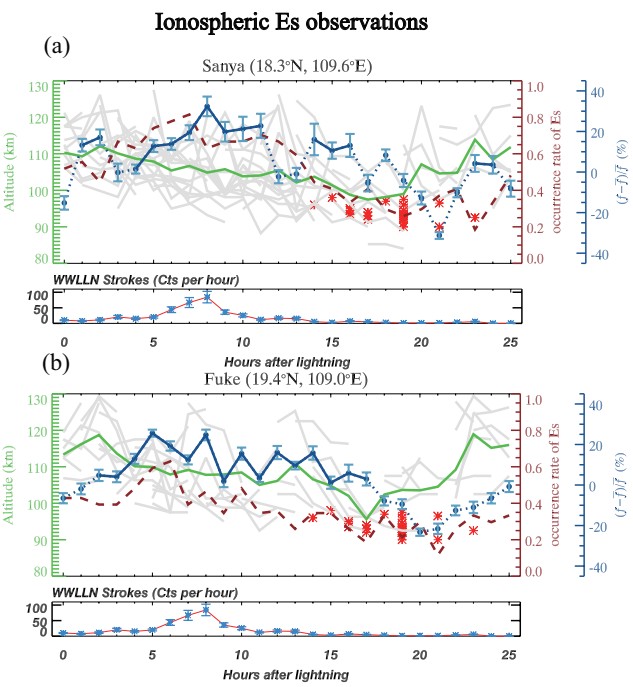

**Figure 2.** (a and b) The ionospheric observations by two digisondes at Sanya (18.3°N, 109.6°E) and Fuke (19.4°N, 109.0°E). All the time series of hourly ionospheric $E_s$ layer are plotted in grey lines, while the green solid line, red dashed line and blue (dotted and solid) lines correspond to the average height, occurrence rate and relative (negative and positive) change of $f_o E_s$ $((f - \overline{f})/\overline{f})$. The lower panels are the averaged WWLLN strokes per hour.





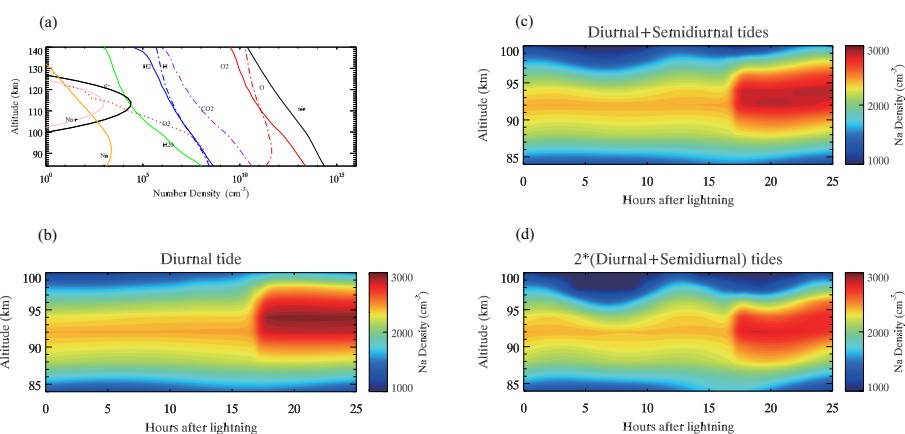

**Figure 3.** The Na chemical simulations (a) The background of concentrations of chemical species and simulated density profiles of Na and Na$^+$ at t=0 h. The simulation results from the Na model with (b) the diurnal tide, (c) diurnal+semidiurnal tides and (d) 2∗(diurnal+semidiurnal) tides.





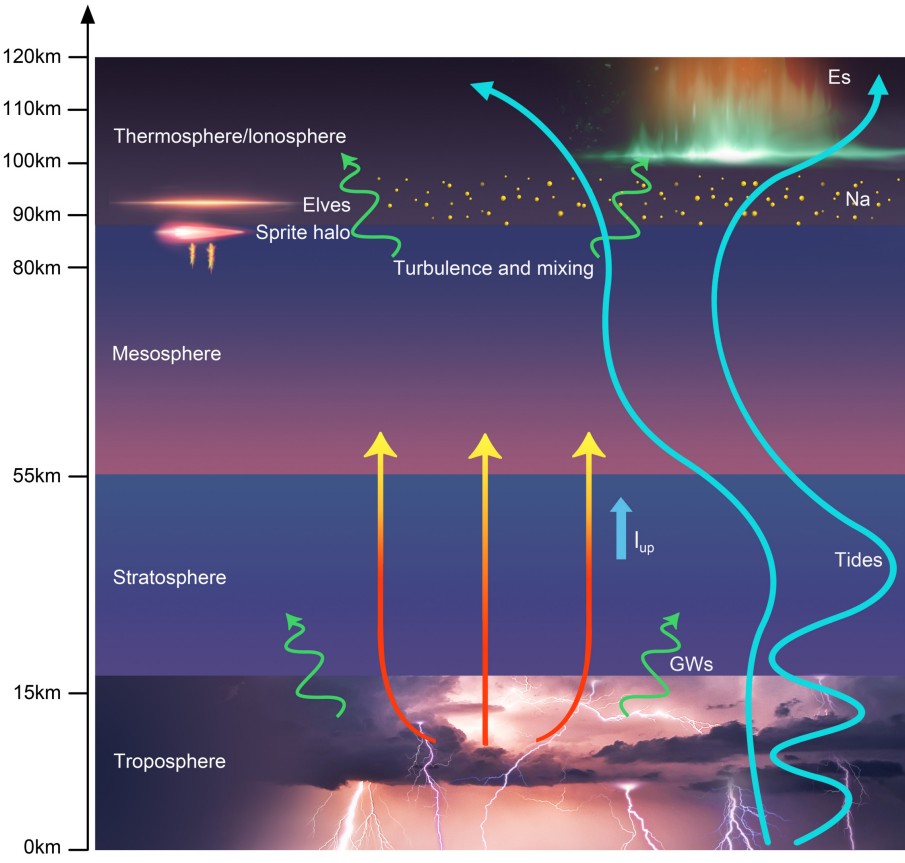

**Figure 4.** Schematic diagram illustrating the proposed mechanism for the lightning-associated enhancement of $E_s$ and neutral Na layer during thunderstorms mainly modulated by atmospheric tides, and potentially influenced by the thunderstorm electrical effects and the tropospheric thunderstorms induced gravity waves (GWs).