# Peer review of "The intensification of metallic layered phenomena above thunderstorms through the modulation of atmospheric tides"

_Atmospheric Chemistry and Physics, 2018_

## Referee Comment (RC1) · Anonymous Referee #1 · 22 Dec 2018

The authors reported a multi-instrument experiment to study the effects of tropospheric thunderstorms on Sporadic E layer activity and on the sodium (Na) layers at the the mesopause region. Furthermore, an Na chemistry model was used to simulate the dynamical and chemical coupling processes in the mesosphere and ionosphere above thunderstorms. The topic is relevant since the exact coupling mechanisms between sporadic E layers and the underlying thunderstorms is still an open question. The study is interesting and a lot of effort have been made to disclose the thunderstorms related processes that act on the Es and Na layers. However, the deduced conclusions from the analysis are too strong. I suggest to write them more carefully.

[Figure]

Thus, I suggest to answer the following questions and comments before acceptance of the manuscript to publish.

Comments in connection with introduction:

1/ I miss a very important review paper of the topic from the introduction which has been published by Haldoupis in 2018: Haldoupis, C. (2018). Is there a conclusive evidence on lightning-related effects on sporadic E layers?. Journal of Atmospheric and Solar-Terrestrial Physics, 172, 117-121.

2/ Page 2. line 15.: Barta et al. 2015 performed Superposed Epoch Analysis (SEA) Barta, V., Pietrella, M., Scotto, C., Bencze, P., Sátori, G., (2015) Thunderstorm-related variations in the sporadic E layer around Rome, Acta Geodaetica et Geophysica, 50:261–270 However, Barta et al. 2017 reported two case studies which was based on highly sampled ionosonde data, Doppler measurement, and ligthning and TLE records. Thus it was a different type of analysis than the previously reported studies, which based on Superposed Epoch Analysis. Please, write it in the introduction more carefully. Barta, V., Haldoupis, C., Sátori, G., Buresova, D., Chum, J., Pozoga, M., Berényi, K. A., Bór,J., Popek, M., Kis, A. and Bencze, P. (2017). Searching for effects caused by thunderstorms in midlatitude sporadic E layers. Journal of Atmospheric and Solar-Terrestrial Physics, 161, 150-159.

Comments to the section 2, Data and Observation

3/ Page 3, line 10 How did you determine the 0 time for SEA? What is the number of lightning at 0 time of the SEA? I suggest to plot the lightning distribution versus time on Fig. 1. as well.

4/ Page 3, line 15 There is a strong enhancement right before the time of lightning.

5/ Page 3, line 15 Generally, the 85-83 km height is too low to observe the Es activity by ionosondes. Comments to Fig. 2. and to the part between line 19 and 30.

6/ What does it mean that hours after lightning? How do you determine the 0 time?

7/ Which territory was take into account for the analysis? How did you estimate the size of this area?

8/ The occurrence rate of Es is the average occurrence of Es during the 28 nights? Please, define the occurrence rate more correctly. What does it show the error bar at the occurrence rate on Fig 2.?

9/ $\hat{f}-$ is the averaged background frequency of foEs, but how did you define exactly? Which time period did you take? Did you take into account the seasonal variation of Es during averaging?

10/ Comment to Fig. 2. In caption: All the time series of the 28 nights? Please, define it more carefully.

11/ Page 3, line 20: misspell: . . .per hour as an indicator

12/ Page 3 Line 33. I can not see good agreement between Davis and Johnson 2005 and this study. In this study it seems that the foEs and Es occurrence rate increasing with the thunderstorm/lightning activity. There is no 6 or 30 hours time delay between the thunderstorm activity and the response of the Es like in the case of Davis and Johnson 2005.

13/ Page 3 line 33: "Both the occurrence rate and relative change of Es vary with the development of the underlying thunderstorm." Please write it more carefully, e.g. Both the occurrence rate and relative change of Es seems to vary with the development of the underlying thunderstorm.

14/ Page 4 line 1: "foEs=5.03 MHz" what is this value? The relative foEs at the peak?

15/ Page 4 line 8-10: Please, write the text more carefully.

16/ Page 4 line 9-10: "the DGS-256 digisonde ionograms from Fuke were automatically scaled" The automatic scaling of foEs parameter is not reliable. I suggest to check the ionograms manually. Especially, because the occurrence rate of Es measured at Fuke

reaches its maximum before the peak of the lightning activity, which is not consistent with the previous states.

Comments in connection with section 3, chemical simulation:

17/ Page 5 line 15-18: "then the ionospheric observations of digisonde at Fuke, near the Na lidar, from t=0 to 25 h are input to drive the Na model." The ionograms measured at Fuke have not been manually checked, so I suggest to check them manually before use them as an input of the model.

18/ Page 5 line 18-19: "Profiles of e and Na + (4% of metallic ions measured by in-situ rocket flights (Kopp, 1997)) are input according to the ionospheric Es observation." How do you determine the e and especially the Na+ profile from Es observation? Please, write more details about this method.

19/ Page 6 line 15-16: "The intensification of Es layer and Na layer are associated metallic layered phenomena through the modulation of atmospheric tides during thunderstorms." Please, write this sentence more carefully, e.g. The intensification of Es layer and Na layer seems to associate with metallic layered phenomena through the modulation of atmospheric tides during thunderstorms.

20/ Page 6. line 17-18: "As mentioned above, the statistical results of various studies using the SEA method have exhibited lightning discharges effect on Es layers." Not directly the lightning, more other processes connected to the thunderstorm activity can affect the Es layer.

21/ Page 6 line 22-23: "Figure 2 shows the similar lag time that the peak of relative foEs occurs 8 h after lightning, comparable to 6 h." Looking at the Fig. 2a. carefully it seems that the time of the Es's occurrence rate is in good agreement with the peak of the lightning activity, there is no 8 h time delay.

Questions and comments related to session 4, Discussion and conclusions:

22/ General comment: Please, try to explain more precisely what you observed and

your proposed coupling mechanisms between the thunderstorm and the Es based on the observations. Please, write more details about the mosulation of tides by GWs. What is the time period necessary for the modulation process? Is it consistent with the time delay reported by Davis and Johnson 2005 or detected in this study?

23/Page 6. line 29.: "hypothesized that the metal atoms may be related to thunderstorms." I can not understand this part of the sentence. Please, write it in a different way.

24/ Page 6. line 30: "It remains possible that the observed lightning-induced ionization enhancement in the Es is associated with the TLEs (Johnson and Davis, 2006)." It is hard to explain the reported 6 and 30 hours time delay of the ionospheric response (Davis and Johnson 2005) by the action of the TLEs.

25/ Page 7. line 11: "The results presented here show robust evidence that the thunderstorm electrical effects accelerate and reinforce this process from metallic Na + ions to neutral Na atoms." It is a very strong state. I can not see any robust evidence based on this study. Please, write it more carefully.

General comment to the figures: The size of the text on the figures should be larger.
* * *

---

## Referee Comment (RC2) · Anonymous Referee #2 · 31 Dec 2018

This is an interesting study which combines and extends two previous works by the first author, one of which linked lightening to Es observations, the 2nd linked lightening to neutral Na observations. Here they combine the three datasets and additionally, use a Na model. It is potentially an intellectual advance over previous efforts and relevant to ACP readers. My main concern, and it is a substantive one, is that the model results are very inadequate and not convincing. They do not yet overcome the skepticism that has been presented in the literature (which was not cited by the authors) concerning the reality of these kinds of observations. My overall evaluation of this work is that it holds promise, but publication is premature.

[Figure]

To show that the enhancement is due to tides, there are two things I believe are needed-both require a significantly expanded modeling/diagnostic effort.

First, I would first need to see a case without tides for comparison. Then a difference field to show the effect. As it stands Figures 3b-d look too similar to discern substantive differences. If anything, it appears that Figure 3d shows a weaker effect. Thus Line 7 on page 6 seems to be incorrect. In which case, this would argue against the proposed mechanism. If they can better document that tides do cause the Na enhancement, then they need to show what term in their complicated equation is the determining factor. If it is vertical wind, then they need to compare that amongst their various models. Finally, how does this Na variation link lightening to electron density or Na+? They say on page 4 that their model calculates Na+ and e-. In which case, since the ultimate goal of this exercise is to show sporadic E, they need to present the variation of these charged species with respect to the differing tidal inputs.

The second overall obstacle to be overcome are the arguments of Haldoupis in his 2018 review in J.Atm Solar Terr Phys., vol 172, pp 117-121. This is a paper which is not cited by the authors, but needs to be addressed. He is skeptical of the causal link between lightening and Es and a key reason is the time delay. Quoting the authors' previous work he questions what mechanism could produce such an enhancement after such long time delays (34 hours for Yu et al, 2015; 19 hours for Yu et al., 2017). Now I recognize that Haldoupis is not addressing tides, but rather gravity waves. Nonetheless I believe his criticisms are relevant here. Specifically, what about their model produces the Na enhancements at the indicated time? Why 17-18 hours? Superficially, this enhancement time does appear somewhat close to the 19 hour sodium enhancement seen in Yu 2017. But then how does this relate to the 34 hour enhancement? Haldoupis is very concerned that the reported time scales lack physical significance. I share those concerns and believe that the authors need to respond to these arguments.

Other concerns

1. I am confused as to what is the input to versus output from their model. They state on page 4, line 20, that "Na+ and e- are calculated from the solution of ...." But then on page 5 lines 18-19, that "profiles ... are input...". I do not understand how these can both be simultaneously true.

2. Their discussion section does not convince me that tides are the root cause. In other words, why should tides, a global resonance phenomenon, vary due to localized thunderstorm activity over East Asia? Reading the discussion, I'm more inclined to think they are arguing for gravity waves; however, Haldoupis has already rejected these arguments.

3. I do not completely understand Figure 2. For the top panels, what are the grey lines- what are the units? They have 4 colors plotted, but only 3 axes.

4. For the bottom panel, they show lightening strokes. But the axis says "hours after lightning". How is this defined? What is time zero?

5. What local time corresponds to t=0 in their model? I am presuming they are doing a fully diurnally resolved calculation (otherwise, its validity would be questionable). So does the local matter as an uncertainty? What happens if they change it?

6. I do not see how Figure 4 helps to understand their mechanism. It's a cartoon- little more. Either enhance the figure (and the argument) or delete it.

7. I know this data has been presented before but I am unable to find out how the 197 nights of lightening data are distributed with respect to season. Can they elaborate? They mention the month of July on page 5 line 12. Certainly sporadic E has a strong seasonality. I would expect, as part of model validation, that they demonstrate that they can reproduce this seasonality.

---

## Author Comment (AC1) · 11 Mar 2019

We would like to thank the reviewers for their valuable comments and suggestions. We have studied all comments carefully and these comments have helped us to significantly improve our manuscript. Following the reviewers' comments, we revised the manuscript. Our responses to the reviewers' comments and corresponding changes with page and line numbers in the revised manuscript are both detailed below in blue text. We mark the major changes in the track-change manuscript.

Reviewer #1 comments (RC1): The authors reported a multi-instrument experiment to study the effects of tropospheric thunderstorms on Sporadic E layer activity and on the

sodium (Na) layers at the mesopause region. Furthermore, an Na chemistry model was used to simulate the dynamical and chemical coupling processes in the mesosphere and ionosphere above thunderstorms. The topic is relevant since the exact coupling mechanisms between sporadic E layers and the underlying thunderstorms is still an open question. The study is interesting and a lot of effort have been made to disclose the thunderstorms related processes that act on the Es and Na layers. However, the deduced conclusions from the analysis are too strong. I suggest to write them more carefully.

Response: Thank you for your positive comments. In the revised manuscript, according to the reviewer's comments, we have revised the manuscript and written them more carefully.

Thus, I suggest to answer the following questions and comments before acceptance of the manuscript to publish.

Comments in connection with introduction: 1/ I miss a very important review paper of the topic from the introduction which has been published by Haldoupis in 2018: Haldoupis, C. (2018). Is there a conclusive evidence on lightning-related effects on sporadic E layers?. Journal of Atmospheric and Solar-Terrestrial Physics, 172, 117-121.

Response: Thanks for your comments. We have read and cited this review paper. Haldoupis, (2018) is concerned about the very large time delay between lightning and Es layers of more than 30 hours (e.g., 6 hours and 30 hours after lightning in Davis and Johnson, (2005)). In this study, we are trying to explain the large time delay, combined with observations of Es and Na layers as well as the chemical simulation.

The time delay of 6 hours may be a result of different definition of the thunderstorm activity (by hourly lightning event in the SEA or the lightning stroke rate actually). In Figure 2, both the occurrence rate and relative change of Es seem to vary with the development of the underlying thunderstorms, when using the average rate of WWLLN

strokes as an indicator of the intensity of thunderstorms. The foEs reaches its peak after 8 hours after the beginning of thunderstorms (the trigger time in the SEA), which is close to the time delay of 6 hours between the thunderstorm activity and the response of the Es in the case of Davis and Johnson (2005).

The time delay of 30 hours in previous statistical studies could be explained by the tidal periodicities in the Es variability (Haldoupis, 2012). In our case study, we show that the height and critical frequency variations of the Es layers are modulated by atmospheric tides. In the Na chemical simulation, we give several cases of the variation of Na density with modulation of a) No tides, b) Semidiurnal tide, c) Diurnal tide and d) both Diurnal and Semidiurnal tides. The case of diurnal and semidiurnal tides is consistent with the Na lidar observations. In fact, the 24-h diurnal tidal modulation can be found in the very large time lags observed between the response of Es layer and lighting, e.g., Es intensifications were observed to occur ∼6 h and ∼30 h after lightning (Davis and Johnson, 2005). The time delay of a subsequent enhancement in foEs at 30 hours after lightning is 24 hours after the enhancement of 6 hours. The tidal periodicities are known to dominate the Es variability (Haldoupis, 2018). Therefore, effects of lightning on the Es layers are observed to occur many hours after lightning as a result from the modulation of diurnal and semidiurnal tides. Besides, the gravity waves, as well as nonlinear GW breaking effects can contribute to the Es variability in the time scale of several hours (Haldoupis, 2018).

Changes: Please see page 2 lines 27-28. "Haldoupis (2018) is concerned about the very large time delay between lightning and Es layers of more than 30 hours (e.g., 6 hours and 30 hours after lightning in Davis and Johnson (2005))."

Please see page 4 lines 21-25. "The foEs reaches its peak 8 hours after the beginning of thunderstorms (the trigger time in the SEA), which is close to the time delay of 6 hours between the thunderstorm activity and the response of the Es in the case of Davis and Johnson (2005). The time delay of 6 hours may be a result of different definition of the thunderstorm activity (by hourly lightning event in the SEA or the lightning

stroke rate actually)."

Please see page 7 lines 30- page 8 lines 4. "The lightning-induced response of Es layers is found to occur several hours after lightning, which makes it difficult to explain these phenomena by thunderstorms generated GWs or lightning-emitted electromagnetic pulses. In fact, the very large time delay in previous statistical studies could be explained by the tidal periodicities in the Es variability. A 24-h diurnal tidal modulation could be found in the very large lag times observed between the response of Es layer and lighting in previous statistical study, e.g., Es intensifications were observed to occur ∼6 h and ∼30 h after lightning (Davis and Johnson, 2005). In this paper, both the occurrence rate and foEs of Es seem to vary with the development of the underlying thunderstorms, when we consider the average rate of WWLLN strokes as an indicator of the intensity of thunderstorms. Figure 2 shows the similar lag time that the peak of relative foEs occurs ∼8 h after lightning trigger time, comparable to ∼6 h in Davis and Johnson (2005). The atmospheric tide is a dominant dynamical process in the MLT and plays dominant roles in the formation and dynamical processes of the intensification of metallic layered phenomena above thunderstorms."

2/ Page 2. line 15.: Barta et al. 2015 performed Superposed Epoch Analysis (SEA) Barta, V., Pietrella, M., Scotto, C., Bencze, P., Sátori, G., (2015) Thunderstorm-related variations in the sporadic E layer around Rome, Acta Geodaetica et Geophysica, 50:261–270 However, Barta et al. 2017 reported two case studies which was based on highly sampled ionosonde data, Doppler measurement, and ligthning and TLE records. Thus it was a different type of analysis than the previously reported studies, which based on Superposed Epoch Analysis. Please, write it in the introduction more carefully. Barta, V., Haldoupis, C., Sátori, G., Buresova, D., Chum, J., Pozoga, M., Berényi, K. A., Bór,J., Popek, M., Kis, A. and Bencze, P. (2017). Searching for effects caused by thunderstorms in midlatitude sporadic E layers. Journal of Atmospheric and Solar-Terrestrial Physics, 161, 150-159.

Response: Thanks for your comments. We have written the ariticles of Barta et al.,

(2015) and Barta et al. (2017) in the introduction more carefully.

Changes: Please see page 2 lines 15-17. "After that, more studies applied the same methodology of SEA on lightning and Es layer measurements using the hourly lightning events as trigger times (Johnson and Davis, 2006; Davis and Lo, 2008; Barta et al., 2013, 2015; Yu et al., 2015)."

Please see page 2 lines 24-26. "Recently, Barta et al. (2017) reported two case studies of mesoscale convective storms that moved through two ionosonde stations, in which a reduction and then disappearance of the ongoing Es layer were observed. It is a non-statistical study, based on highly sampled ionosonde data, Doppler measurement, and lightning and TLE records."

Comments to the section 2, Data and Observation 3/ Page 3, line 10 How did you determine the 0 time for SEA? What is the number of lightning at 0 time of the SEA? I suggest to plot the lightning distribution versus time on Fig. 1. as well.

Response: Thanks for your comments. We have explained it in more details in the revised manuscript. In the SEA, to analyze such a large data set, the lightning data from WWLLN and observations by the Na resonance fluorescence lidar at Haikou were sorted to hourly data to investigate the connection between thunderstorms and neutral metal Na layer (Yu et al., 2017). By aligning the Na density data according to the time of the lightning event ("trigger" time) and calculating the average response of all the Na layer profiles that were acquired $\Delta t$ hours either before and after lightning, any consistent influence of lightning and thunderstorms on the Na layer will be reinforced while other random influences unassociated with lightning will be canceled out. The number of triggers is plotted in Figure 1 as Yu et al. (2017). The 0 time is the trigger time in the SEA. However, the hourly number of trigger time cannot exhibit the development of thunderstorms at different stage. In this study, we consider the averaged WWLLN strokes per hour as an indicator of the intensity of thunderstorms as shown in Figure 2.

The distribution of lightning trigger events is plotted in Figure 1, and distribution of

lightning stroke rate is plotted in Figure 2.

Changes: Please see page 3 lines 14-24 and changes in Figure 1. "In the SEA, to analyze such a large data set, the lightning data from WWLLN and observations by the Na resonance fluorescence lidar were sorted to hourly data to investigate the connection between thunderstorms and neutral metal Na layer...In this case study, the 0 time is the trigger time in the SEA and the number of triggers is plotted."

4/ Page 3, line 15 There is a strong enhancement right before the time of lightning.

Response: Thanks for your comments. In the SEA, some undesired recurrence of lightning events before time zero would contaminate the result (Yu et al., 2017). But on the whole, the enhancement of Na density at altitudes of 85–98 km exceeds the significant level at 17–24 h, 85–98 km after lightning with the maximum increase of ∼600 cm-3 at t=19 h, at an altitude of 93 km.

Changes: Please see page 3 lines 26-29. "Some undesired recurrence of lightning events before time zero would contaminate the result (Yu et al., 2017). The Na layer is obviously intensified up to 500 cm-3 after lightning. The zoom-in window between 0 and 25 h plotted in red is shown in Figure 1b. The enhancement of Na layer is evident at 17-24 h, 85-98 km after lightning with the maximum increase of ∼600 cm-3 at t=19 h, at an altitude of 93 km."

5/ Page 3, line 15 Generally, the 85-83 km height is too low to observe the Es activity by ionosondes. Comments to Fig. 2. and to the part between line 19 and 30.

Response: Thanks for your comments. Generally, Es layers could not be formed by the wind shear theory below 90 km because of the high ion-neutral collision frequencies. There is only one case out of 28 night ionosonde observations. This case existed in two hourly profiles. The digisonde ionograms were manually scaled.

6/ What does it mean that hours after lightning? How do you determine the 0 time? Response: The 0 time is the trigger events for SEA. We further analyzed the cases

distributed on 28 thunderstorms in Yu et al. (2017). In the case study, the averaged lightning stroke rate was used as an indicator of the intensity of thunderstorms at the beginning of triggers.

Changes: Please see page 3 lines 14-24. "In the SEA, to analyze such a large data set, the lightning data from WWLLN and observations by the Na resonance fluorescence lidar were sorted to hourly data to investigate the connection between thunderstorms and neutral metal Na layer...In this case study, the 0 time is the trigger time in the SEA and the number of triggers is plotted."

7/ Which territory was take into account for the analysis? How did you estimate the size of this area?

Response: The territory is thin $\pm 1°$ latitude and longitude of Na lidar location, as described in (Yu et al., 2017). The previous statistical studies show that the lightning activity has a significant effect on the ionosphere within âĹij100 km (Johnson and Davis, 2006, Yu et al., 2015).

8/ The occurrence rate of Es is the average occurrence of Es during the 28 nights? Please, define the occurrence rate more correctly. What does it show the error bar at the occurrence rate on Fig 2.? Response: Thanks for your comments. The occurrence rate of Es is the average occurrence of Es during the 28 nights. The error bar at the relative change of foEs is the standard error of relative change of foEs during the 28 nights. We have defined these in the revised manuscript more correctly.

Changes: Please see page 4 lines 9-15. "The average occurrence rate of Es during the 28 nights is plotted in red dashed lines...The error bar at the relative change of frequency is the standard error of relative change of foEs during the 28 nights."

9/ fËĘ– is the averaged background frequency of foEs, but how did you define exactly? Which time period did you take? Did you take into account the seasonal variation of Es during averaging?

Response: Thanks for your comments. is the averaged background frequency from 2010 to 2013. To minimize the effects of the height on the frequency, the averaged background frequency is defined as the mean frequency of the dataset within ±0.5 km of the Es height at each point. Because most of cases are distributed in summer and the study period is one day, we take the influences of the height on the frequency into account during averaging, instead of the seasonal variation of Es.

10/ Comment to Fig. 2. In caption: All the time series of the 28 nights? Please, define it more carefully.

Response: Thanks. We have defined it more carefully. Changes: Please see the caption of Figure 2 in page 16. "All the time series of hourly profiles of Es height during the 28 nights are plotted in grey lines, while the green solid lines, red dashed lines and blue (dotted and solid) lines correspond to the average height, occurrence rate and relative (negative and positive) change of foEs ."

11/ Page 3, line 20: misspell: . . .per hour as an indicator

Response: Thanks. We have corrected it. Changes: Please see page 4 lines 1-2. "The lower panel is the average number of WWLLN strokes per hour as an indicator of the intensity of thunderstorms."

12/ Page 3 Line 33. I can not see good agreement between Davis and Johnson 2005 and this study. In this study it seems that the foEs and Es occurrence rate increasing with the thunderstorm/lightning activity. There is no 6 or 30 hours time delay between the thunderstorm activity and the response of the Es like in the case of Davis and Johnson 2005.

Response: Thanks for your comments. In the previous SEA studies (Davis and Johnson, 2005; Johnson and Davis, 2006; Davis and Lo, 2008; Barta et al., 2013, 2015; Yu et al., 2015; Yu et al., 2017), the hourly lightning data can hardly exhibit the development of thunderstorms at different stage. In this study, when using the averaged

WWLLN stroke rate as an indicator of the intensity of thunderstorms, it seems that the foEs and Es occurrence rate increase with the development of thunderstorm activity, as shown in Figure 2. The foEs reaches its peak 8 hours after the beginning of thunderstorms (the trigger events in the SEA), which is consistent with the time delay of 6 hours between the thunderstorm activity and the response of the Es in the case of Davis and Johnson (2005). The time delay of 6 hours may be a result of different definition of the thunderstorm activity (by hourly lightning event in the SEA or the lightning stroke rate actually). Besides, the gravity waves generated by thunderstorm and lightning, as well as nonlinear GW breaking effects can contribute to the Es variability in the time scale of several hours (Haldoupis, 2018).

Among cases of our simulations, the case of diurnal and semidiurnal tides is consistent with the Na lidar observations, indicating a modulating action of diurnal and semidiurnal tides on the intensification of metallic layered phenomena above thunderstorms. The 24-h diurnal tidal modulation can be found in the very large time lags observed between the response of Es layer and lighting, e.g., Es intensifications were observed to occur ∼6 h and ∼30 h after lightning (Davis and Johnson, 2005). The time delay of a subsequent enhancement in foEs at 30 hours after lightning is 24 hours after the enhancement of 6 hours.

Changes: Please see page 4 lines 21-25. "The foEs reaches its peak 8 hours after the beginning of thunderstorms (the trigger time in the SEA), which is close to the time delay of 6 hours between the thunderstorm activity and the response of the Es in the case of Davis and Johnson (2005). The time delay of 6 hours may be a result of different definition of the thunderstorm activity (by hourly lightning event in the SEA or the lightning stroke rate actually)."

Please see page 7 lines 30- page 8 lines 4. "The lightning-induced response of Es layers is found to occur several to many hours after lightning, which makes it difficult to explain these phenomena by thunderstorms generated GWs or lightning-emitted electromagnetic pulses. In fact, the very large time delay in previous statistical studies

could be explained by the tidal periodicities in the Es variability. A 24-h diurnal tidal modulation could be found in the very large lag times observed between the response of Es layer and lighting in previous statistical study, e.g., Es intensifications were observed to occur ∼6 h and ∼30 h after lightning (Davis and Johnson, 2005). In this paper, both the occurrence rate and foEs of Es seem to vary with the development of the underlying thunderstorms, when we consider the average rate of WWLLN strokes as an indicator of the intensity of thunderstorms. Figure 2 shows the similar lag time that the peak of relative foEs occurs ∼8 h after lightning trigger time, comparable to ∼6 h in Davis and Johnson (2005). The atmospheric tide is a dominant dynamical process in the MLT and plays dominant roles in the formation and dynamical processes of the intensification of metallic layered phenomena above thunderstorms."

13/ Page 3 line 33: "Both the occurrence rate and relative change of Es vary with the development of the underlying thunderstorm." Please write it more carefully, e.g. Both the occurrence rate and relative change of Es seems to vary with the development of the underlying thunderstorm.

Response: Thanks for your comments. We have written it more carefully. Changes: Please see page 4 lines 16-17. "Both the occurrence rate and relative change of Es seem to vary with the development of the underlying thunderstorms, when using the average rate of WWLLN strokes as an indicator of the intensity of thunderstorms."

14/ Page 4 line 1: "foEs=5.03 MHz" what is this value? The relative foEs at the peak?

Response: Yes, it is the relative foEs at the peak. Changes: Please see page 4 lines 20. "the average relative foEs at the peak is 5.03 MHz"

15/ Page 4 line 8-10: Please, write the text more carefully.

Response: We have written the text more carefully. Changes: Please see page 4 lines 31-33. "Figure 2b shows data from another digisonde at Fuke near the Na lidar. The foEs increases with the thunderstorm activity. The peak of the relative foEs reaches

30–35% (the average relative foEs at the peak is 4.9–5.0 MHz) and the maximum of occurrence rate is ∼0.6."

16/ Page 4 line 9-10: "the DGS-256 digisonde ionograms from Fuke were automatically scaled" The automatic scaling of foEs parameter is not reliable. I suggest to check the ionograms manually. Especially, because the occurrence rate of Es measured at Fuke reaches its maximum before the peak of the lightning activity, which is not consistent with the previous states.

Response: Thanks for your comments. Following your suggestion, we manually scaled the digisonde ionograms from Fuke. The ionosonde data at Fuke were manually scaled and then the foEs and the occurrence rate of Es are shown in Figure 2b. The foEs reaches its maximum during the peak of the lightning activity. The occurrence rate of Es measured at Fuke reaches its maximum before the peak of the lightning activity. It could be likely a result of the combined effects on the occurrence rate of Es, with a development of thunderstorms and a decrease in height of Es.

Changes: Please see page 4 lines 33 - page 5 lines 3. "The occurrence rate of Es measured at Fuke reaches its maximum before the peak of the thunderstorm activity. It is likely a result of the competitive effects on the occurrence rate of Es, with a development of thunderstorms and a decrease in height of Es. It shows the similar tendency of tide-period height. With the decrease in the Es height below 100 km and the occurrence of the maximum neutral Na layer during thunderstorms, a remarkable reduction of relative foEs and then disappearance of the Es with a decrease in occurrence rate are observed, as reported in Barta et al. (2017). Furthermore, Figure 2b shows the Es height is controlled not only by the diurnal tide but also by the semidiurnal tide."

Comments in connection with section 3, chemical simulation: 17/ Page 5 line 15-18: "then the ionospheric observations of digisonde at Fuke, near the Na lidar, from t=0 to 25 h are input to drive the Na model." The ionograms measured at Fuke have not been manually checked, so I suggest to check them manually before use them as an input

of the model.

Response: Thanks for your comments. Following your suggestion, we manually scaled the digisonde ionograms from Fuke during the period of 2010-2013.

Changes: Please see the changes in Figure 2 and 3.

18/ Page 5 line 18-19: "Profiles of e and Na + (4% of metallic ions measured by in-situ rocket flights (Kopp, 1997)) are input according to the ionospheric Es observation." How do you determine the e and especially the Na+ profile from Es observation? Please, write more details about this method.

Response: Rocket-borne mass spectrometric measurements proved that the Es layer is mostly the ionization of metal atoms such as $Fe+$, $Mg+$, and $Na+$ (Kopp, 1997; Grebowsky and Aikin, 2002). 4% of metallic ions measured are the $Na+$ ions, by in-situ rocket flights (Kopp, 1997). Therefore, the electron concentration of the Es, Ne, is estimated by the formula , where Ne is in /cm-3 and f is the critical frequency in MHz.

Changes: Please see page 6 lines 18-21. "Rocket-borne mass spectrometric measurements have proved that the Es layer is mostly the ionization of metal atoms, in which ∼4% of ions measured are the $Na+$ ions (Kopp, 1997; Grebowsky and Aikin, 2002). The electron concentration of the Es layer in cm-3 is estimated from foEs in MHz, by the formula .

19/ Page 6 line 15-16: "The intensification of Es layer and Na layer are associated metallic layered phenomena through the modulation of atmospheric tides during thunderstorms." Please, write this sentence more carefully, e.g. The intensification of Es layer and Na layer seems to associate with metallic layered phenomena through the modulation of atmospheric tides during thunderstorms.

Response: Thanks for your comments. We have written it more carefully. Changes: Please see page 7 lines 27-28. "The intensification of Es layer and Na layer seems to associate with metallic layered phenomena through the modulation of atmospheric

tides during thunderstorms."

20/ Page 6. line 17-18: "As mentioned above, the statistical results of various studies using the SEA method have exhibited lightning discharges effect on Es layers." Not directly the lightning, more other processes connected to the thunderstorm activity can affect the Es layer. Response: Thanks for your comments. We have revised it. Changes: Please see page 7 lines 28-30. "As mentioned above, the statistical results of various studies using the SEA method have exhibited processes connected to the thunderstorm activity can affect Es layers"

21/ Page 6 line 22-23: "Figure 2 shows the similar lag time that the peak of relative foEs occurs 8 h after lightning, comparable to 6 h." Looking at the Fig. 2a. carefully it seems that the time of the Es's occurrence rate is in good agreement with the peak of the lightning activity, there is no 8 h time delay.

Response: Thanks for your comments.

The hourly number of trigger time in the SEA cannot exhibit the development of thunderstorms at different stage. In this study, we consider the averaged WWLLN strokes per hour as an indicator of the intensity of thunderstorms at the beginning of triggers, as shown in Figure 2. The time 0 is the time of trigger events (at the beginning of thunderstorms) for 28 nights in (Yu et al., 2017). Figure 2 shows the lag time that the peak of relative foEs occurs 7-9 h after the beginning of thunderstorms. The previous statistical studies show that the Es intensifications were observed to occur $\sim$6 h and $\sim$30 h after lightning (trigger events) (Davis and Johnson, 2005). Therefore, the lag time of $\sim$6 hours is comparable to 7-9 hours in this study. It is a result of different definition of the thunderstorm activity (by hourly lightning event in the SEA or the lightning stroke rate actually).

Changes: Please see page 4 lines 21-25. "The foEs reaches its peak 8 hours after the beginning of thunderstorms (the trigger time in the SEA), which is close to the time delay of 6 hours between the thunderstorm activity and the response of the Es in

the case of Davis and Johnson (2005). The time delay of 6 hours may be a result of different definition of the thunderstorm activity (by hourly lightning event in the SEA or the lightning stroke rate actually)."

Questions and comments related to session 4, Discussion and conclusions: 22/ General comment: Please, try to explain more precisely what you observed and your proposed coupling mechanisms between the thunderstorm and the Es based on the observations. Please, write more details about the modulation of tides by GWs. What is the time period necessary for the modulation process? Is it consistent with the time delay reported by Davis and Johnson 2005 or detected in this study?

Response: Thanks for your comments. We have explained more precisely that we observed and the proposed coupling mechanisms. The enhancement of Es layer above thunderstorms is related to the subsequent enhancement of metal Na layer. The large lag time is associated to the modulation of atmospheric tides. In the Na chemical simulations, we added more cases of the variation of Na density with modulation of a) no tides, b) semidiurnal tide, c) diurnal tide and d) both diurnal and semidiurnal tides. The case of diurnal and semidiurnal tides is consistent with the Na lidar observations. The result indicates that the modulation of atmospheric tides play an important role in the time delay of the response of Es layer and Na layer.

Changes: Please see page 7 lines 30- page 8 lines 5. "The lightning-induced response of Es layers is found to occur several hours after lightning, which makes it difficult to explain these phenomena by thunderstorms generated GWs or lightning-emitted electromagnetic pulses. In fact, the very large time delay in previous statistical studies could be explained by the tidal periodicities in the Es variability. A 24-h diurnal tidal modulation could be found in the very large lag times observed between the response of Es layer and lighting in previous statistical study, e.g., Es intensifications were observed to occur ∼6 h and ∼30 h after lightning (Davis and Johnson, 2005). In this paper, both the occurrence rate and foEs of Es seem to vary with the development of the underlying thunderstorms, when we consider the average rate of WWLLN strokes

as an indicator of the intensity of thunderstorms. Figure 2 shows the similar lag time that the peak of relative foEs occurs ∼8 h after lightning trigger time, comparable to ∼6 h in Davis and Johnson (2005). The atmospheric tide is a dominant dynamical process in the MLT and plays dominant roles in the formation and dynamical processes of the intensification of metallic layered phenomena above thunderstorms"

23/Page 6. line 29.: "hypothesized that the metal atoms may be related to thunderstorms." I can not understand this part of the sentence. Please, write it in a different way.

Response: Thanks for your comments. We have changed as 'proposed that the variation in metal atoms may be related to lightning-associated intensification of Es layer above thunderstorms' Changes: Please see page 8 lines 9-10. "proposed that the variation in metal atoms may be related to lightning-associated intensification of Es layer above thunderstorms."

24/ Page 6. line 30: "It remains possible that the observed lightning-induced ionization enhancement in the Es is associated with the TLEs (Johnson and Davis, 2006)." It is hard to explain the reported 6 and 30 hours time delay of the ionospheric response (Davis and Johnson 2005) by the action of the TLEs.

Response: Thanks for your comments. In this study, we have explained the reported 6 and 30 hours time delay of the ionospheric response could be a result of different definition of the thunderstorm activity (by hourly lightning event in SEA or the lightning stroke rate), and the modulation of semidiurnal/ diurnal tides. In fact, both the occurrence rate and relative change of Es seems to vary with the development of the underlying thunderstorm. Therefore, the electrodynamic coupling between lightning and ionosphere should not be totally excluded (Haldoupis, 2018). The actions of gravity waves generated by the underlying thunderstorms are also one of the possible mechanisms.

25/ Page 7. line 11: "The results presented here show robust evidence that the thunderstorm electrical effects accelerate and reinforce this process from metallic Na + ions

to neutral Na atoms." It is a very strong state. I can not see any robust evidence based on this study. Please, write it more carefully. Response: We have written it more carefully. Changes: Please see page 8 lines 23-25. "The results presented here may show that the thunderstorm electrical effects could accelerate and reinforce this process from metallic Na+ ions to neutral Na atoms"

General comment to the figures: The size of the text on the figures should be larger. Response: Thanks for your comments. We have changed it.

Reviewer #2 comments (RC2):

This is an interesting study which combines and extends two previous works by the first author, one of which linked lightening to Es observations, the 2nd linked lightening to neutral Na observations. Here they combine the three datasets and additionally, use a Na model. It is potentially an intellectual advance over previous efforts and relevant to ACP readers. My main concern, and it is a substantive one, is that the model results are very inadequate and not convincing. They do not yet overcome the skepticism that has been presented in the literature (which was not cited by the authors) concerning the reality of these kinds of observations. My overall evaluation of this work is that it holds promise, but publication is premature.

Response: Thank you for your positive comments. In the revised manuscript, according to the reviewer's comments, we have improved the model results. We consider the variation of Es height with atmospheric tides and give more cases with the Harmonic fitting of corresponding diurnal and semidiurnal tides. In the Na chemical simulation, we give the cases of the variation of Na density with modulation of a) No tides, b) Semidiurnal tide, c) Diurnal tide and d) both Diurnal and Semidiurnal tides. Atmospheric tides control the dynamical process in the MLT region and the Es layer descends with a tidal downward phase. The intensification of Es layer and Na layer associates with metallic layered phenomena through the modulation of atmospheric tides during thunderstorms. Figure 3c and 3d show the electron density of Es layer and the simulated

Na density without modulation of tides. The Es layer maintains at an altitude of ∼106 km without tidal perturbations. There is a weak response of Na density because the Es does not descend with the tidal phase. Other cases show the time delay of the enhancement of Es layer and Na layer is a result of modulation of diurnal and semidiurnal tides. Note that the cases with semidiurnal or diurnal tidal component alone cannot reflect the actual variation in the height of Es observed by digisonde. The downward phase propagation of the node could be ahead of or behind the observed phase. Actually, the Es layer descends with the vertical downward tidal phase, mostly by both diurnal and semidiurnal tides (Whitehead, 1961, 1989; Mathews, 1998; Haldoupis, 2012). Figure 3i and 3j show the exact case with diurnal and semidiurnal tides. The simulated result in Figure 3j is consistent with the observed enhancement of Na layer.

Changes: Please see page 6 lines 25 - page 7 lines 13. "The average variation in height of Es during 28 nights observed by Fuke digisonde is plotted in green lines in Figure 3b. The blue lines are harmonic fit consisting of corresponding duirnal (red lines) and semiduirnal (pink lines) components. Figures 3c–3j are the simulation results from the Na model in consideration of modulation with (1) no tides, (2) semidiurnal tide, (3) diurnal tide, and (4) diurnal+semidiurnal tides...The height of Es is controlled by the diurnal and semidiurnal tides in Figure 3i. The Es layer descends with the vertical tidal phase until it weakens and is depleted below 100 km at t=18 h. The simulated density of Na layer is significantly enhanced with its value of ∼4,200 cm-3 at t=19 h, and at an altitude of 98 km."

To show that the enhancement is due to tides, there are two things I believe are needed both require a significantly expanded modeling/diagnostic effort. First, I would first need to see a case without tides for comparison. Then a difference field to show the effect. As it stands Figures 3b-d look too similar to discern substantive differences. If anything, it appears that Figure 3d shows a weaker effect. Thus Line 7 on page 6 seems to be incorrect. In which case, this would argue against the proposed mechanism. If they can better document that tides do cause the Na enhancement, then they

need to show what term in their complicated equation is the determining factor. If it is vertical wind, then they need to compare that amongst their various models. Finally, how does this Na variation link lightening to electron density or Na+? They say on page 4 that their model calculates Na+ and e-. In which case, since the ultimate goal of this exercise is to show sporadic E, they need to present the variation of these charged species with respect to the differing tidal inputs.

Response: Thank you for your comments. In our case study of observation from two digisondes, we have showed that the height and critical frequency variations of the Es layers are modulated by atmospheric tides. The average variation in height of Es during 28 nights observed by Fuke digisonde is fitted, consisting of corresponding duirnal (red lines) and semiduirnal (pink lines) components in Figure 3b. Then we show simulation results with cases of modulation with (1) no tides, (2) semidiurnal tide, (3) diurnal tide, and (4) diurnal+semidiurnal tides. We present the variation of Es layer and the simulated Na layer.

Figure 3c and 3d show a case without tides for comparison. The Es layer maintains at an altitude of $\sim$106 km without tidal perturbations. There is a weak response of Na density because the Es does not descend with the tidal phase. Other cases show the time delay of the enhancement of Es layer and Na layer is a result of modulation of diurnal and semidiurnal tides.

Among these, the exact case, with modulation of diurnal and semidiurnal tides in Figure 3i and 3j, shows the height of Es is controlled by the diurnal and semidiurnal tides. The Es layer descends with the vertical tidal phase until it weakens and is depleted below 100 km at t=18 h. The simulated density of Na layer occurs at t=19 h, which is consistent with the Na lidar observations.

Changes: Please see the simulation in pages 6 and 7 and corresponding Figure 3.

The second overall obstacle to be overcome are the arguments of Haldoupis in his 2018 review in J.Atm Solar Terr Phys., vol 172, pp 117-121. This is a paper which is

not cited by the authors, but needs to be addressed. He is skeptical of the causal link between lightening and Es and a key reason is the time delay. Quoting the authors' previous work he questions what mechanism could produce such an enhancement after such long time delays (34 hours for Yu et al, 2015; 19 hours for Yu et al., 2017). Now I recognize that Haldoupis is not addressing tides, but rather gravity waves. Nonetheless I believe his criticisms are relevant here. Specifically, what about their model produces the Na enhancements at the indicated time? Why 17-18 hours? Superficially, this enhancement time does appear somewhat close to the 19 hour sodium enhancement seen in Yu 2017. But then how does this relate to the 34 hour enhancement? Haldoupis is very concerned that the reported time scales lack physical significance. I share those concerns and believe that the authors need to respond to these arguments.

Response: Thanks for your comments. We have read and cited this review paper of Haldoupis in J.Atm Solar Terr Phys., vol 172, pp 117-121. Haldoupis, (2018) is concerned about the very large time delay between lightning and Es layers of more than 30 hours (e.g., 6 hours and 30 hours after lightning in Davis and Johnson, (2005)).

The time delay of 6 hours may be a result of different definition of the thunderstorm activity (by hourly lightning event in the SEA or the lightning stroke rate actually). In Figure 2, both the occurrence rate and relative change of Es seem to vary with the development of the underlying thunderstorms, when using the average rate of WWLLN strokes as an indicator of the intensity of thunderstorms. The foEs reaches its peak after 8 hours after the beginning of thunderstorms (the trigger time in the SEA), which is close to the time delay of 6 hours between the thunderstorm activity and the response of the Es in the case of Davis and Johnson (2005).

The time delay of 30 hours in previous statistical studies could be explained by the tidal periodicities in the Es variability (Haldoupis, 2012). In our case study, we show that the height and critical frequency variations of the Es layers are modulated by atmospheric tides. In the Na chemical simulation, we give several cases of the variation of Na density with modulation of a) No tides, b) Semidiurnal tide, c) Diurnal tide and d) both

Diurnal and Semidiurnal tides. The case of diurnal and semidiurnal tides is consistent with the Na lidar observations. In fact, the 24-h diurnal tidal modulation can be found in the very large time lags observed between the response of Es layer and lighting, e.g., Es intensifications were observed to occur ~6 h and ~30 h after lightning (Davis and Johnson, 2005). The time delay of a subsequent enhancement in foEs at 30 hours after lightning is 24 hours after the enhancement of 6 hours. Yu et al. (2015) and Yu et al. (2017) also found an enhancement of Es layer and Na layer after long time delays (34 hours and 19 hours). The tidal periodicities are known to dominate the Es variability (Haldoupis, 2018). Therefore, effects of lightning on the Es layers are observed to occur many hours after lightning as a result from the modulation of diurnal and semidiurnal tides. Besides, the gravity waves, as well as nonlinear GW breaking effects can contribute to the Es variability in the time scale of several hours (Haldoupis, 2018).

Changes: Please see page 2 lines 27-28. "Haldoupis (2018) is concerned about the very large time delay between lightning and Es layers of more than 30 hours (e.g., 6 hours and 30 hours after lightning in Davis and Johnson (2005))"

Please see page 4 lines 21-25. "The foEs reaches its peak 8 hours after the beginning of thunderstorms (the trigger time in the SEA), which is close to the time delay of 6 hours between the thunderstorm activity and the response of the Es in the case of Davis and Johnson (2005). The time delay of 6 hours may be a result of different definition of the thunderstorm activity (by hourly lightning event in the SEA or the lightning stroke rate actually)."

Please see page 7 lines 30- page 8 lines 4. "The lightning-induced response of Es layers is found to occur several to many hours after lightning, which makes it difficult to explain these phenomena by thunderstorms generated GWs or lightning-emitted electromagnetic pulses. In fact, the very large time delay in previous statistical studies could be explained by the tidal periodicities in the Es variability. A 24-h diurnal tidal modulation could be found in the very large lag times observed between the response

of Es layer and lighting in previous statistical study, e.g., Es intensifications were observed to occur ~6 h and ~30 h after lightning (Davis and Johnson, 2005). In this paper, both the occurrence rate and foEs of Es seem to vary with the development of the underlying thunderstorms, when we consider the average rate of WWLLN strokes as an indicator of the intensity of thunderstorms. Figure 2 shows the similar lag time that the peak of relative foEs occurs ~8 h after lightning trigger time, comparable to ~6 h in Davis and Johnson (2005). The atmospheric tide is a dominant dynamical process in the MLT and plays dominant roles in the formation and dynamical processes of the intensification of metallic layered phenomena above thunderstorms."

Other concerns 1. I am confused as to what is the input to versus output from their model. They state on page 4, line 20, that "Na+ and e- are calculated from the solution of . . .." But then on page 5 lines 18-19, that "profiles . . . are input. . .". I do not understand how these can both be simultaneously true.

Response: Sorry for the unclear description. In the model, the Es layer is initialized with a selected percentage of Na+ and then descends at the rate observed by the digisonde. The redistribution and rates of neutralization of Na+ and e-, and changes in the main long-lived Na species, such as Na, NaHCO3 are determined from the solution of continuity equations below, while other short-lived intermediates are considered as steady-state concentration (Plane, 2004).

Changes: Please see page 5 lines 11-14. "In the model, the Es layer is initialized with a selected percentage of Na+ and then descends at the rate observed by the digisonde. The redistribution and rates of neutralization of Na+ and e-, and changes in the main long-lived Na species, such as Na, NaHCO3 are determined from the solution of continuity equations below, while other short-lived intermediates are considered as steady-state concentration (Plane, 2004)"

2. Their discussion section does not convince me that tides are the root cause. In other words, why should tides, a global resonance phenomenon, vary due to localized

thunderstorm activity over East Asia? Reading the discussion, I'm more inclined to think they are arguing for gravity waves; however, Haldoupis has already rejected these arguments.

Response: Thanks for your comments. In Figure 2, both the occurrence rate and relative change of Es seem to vary with the development of the underlying thunderstorms, when using the average rate of WWLLN strokes as an indicator of the intensity of thunderstorms. The mechanism for the enhancement of Es layer and Na layer could be the gravity waves generated during thunderstorms or the electric effect by lightning. However, this study is trying to explain the long time lags of the enhancement. The long time lag could be a result of different definition of the thunderstorm activity (by hourly lightning event in the SEA or the lightning stroke rate actually) and tidal periodicities in the Es variability.

3. I do not completely understand Figure 2. For the top panels, what are the grey lineswhat are the units? They have 4 colors plotted, but only 3 axes. Response: Sorry for the unclear description. The grey lines are all the time series of hourly profiles of Es height during the 28 nights. Other green solid lines, red dashed lines and blue lines correspond to the average height, occurrence rate and relative change of foEs.

Changes: Please see page 4 lines 8-12. "All the time series of hourly profiles of Es height during the 28 nights are plotted in grey lines. The average occurrence rate of Es during the 28 nights is plotted in red dashed lines. The Es layer occurs more frequently during thunderstorms. The relative change of critical frequency, foEs is shown in blue dotted and solid lines, in which is the average background frequency of foEs."

4. For the bottom panel, they show lightening strokes. But the axis says "hours after lightning". How is this defined? What is time zero?

Response: Thanks for your comments. In Figure 1 of the SEA study, the time zero in the statistical study is the lightning event ("trigger" time) in the SEA. However, the hourly number of trigger time in the statistical study cannot exhibit the development of

thunderstorms at different stage. In this study of Figure 2, we consider the average rate of WWLLN strokes as an indicator of the intensity of thunderstorms. Both the occurrence rate and foEs of Es seem to vary with the development of the underlying thunderstorms, when we consider the average rate of WWLLN strokes as an indicator of the intensity of thunderstorms. The time zero is also the time 0 of the trigger time in SEA in Figure 1.

Changes: Please see page 3 lines 14-24 and changes in Figure 1. "In the SEA, to analyze such a large data set, the lightning data from WWLLN and observations by the Na resonance fluorescence lidar were sorted to hourly data to investigate the connection between thunderstorms and neutral metal Na layer. . .In this case study, the 0 time is the trigger time in the SEA and the number of triggers is plotted."

Please see page 3 lines 30-32. "However, the hourly number of trigger time in the statistical study cannot exhibit the development of thunderstorms at different stage. In this study, we consider the average rate of WWLLN strokes as an indicator of the intensity of thunderstorms, as shown in Figure 2"

5. What local time corresponds to t=0 in their model? I am presuming they are doing a fully diurnally resolved calculation (otherwise, its validity would be questionable). So does the local matter as an uncertainty? What happens if they change it? Response: The time is $\sim$ 8 LT based on the average time distributed on 28 thunderstorm nights. Then the monthly averaged diurnal and semidiurnal tides from GSWM are considered in the Na chemistry model. It can also be found that the variation of Es height observed from digisonde in Figure 3 shows a diurnal and semidiurnal tidal modulation. They are both consistent in phase around 100-110 km, and so we treat the average local time on 28 thunderstorm nights as the actual local time.

6. I do not see how Figure 4 helps to understand their mechanism. It's a cartoon- little more. Either enhance the figure (and the argument) or delete it. Response: Thanks for your comments. Figure 4 shows the proposed mechanism in discussion. The

lightning-associated enhancement of Es and neutral Na layer during thunderstorms could be mainly modulated by atmospheric tides, and potentially influenced by the thunderstorm electrical effects and gravity waves. Some part of investigations such as its relationship with TLEs will be a topic of our further studies. 7. I know this data has been presented before but I am unable to find out how the 197 nights of lightening data are distributed with respect to season. Can they elaborate? They mention the month of July on page 5 line 12. Certainly sporadic E has a strong seasonality. I would expect, as part of model validation, that they demonstrate that they can reproduce this seasonality. Response: Thanks for your comments. The 28 nights of lightning data with the Na lidar observation in this study are distributed in primarily in summer (19 out of 28 in June, July and August). Therefore, in the simulation, the simultaneous observation of Es layer and the general condition over Haikou in July are used. We do not study the Es seasonality in simulations as the observed thunderstorm nights are distributed in summer. The average variations of Es height and Es value from digisonde during the 28 nights are used in the simulation.

ReferencesïijŽ Barta, V., Scotto, C., Pietrella, M., Sgrigna, V., Conti, L., and Sátori, G.: A statistical analysis on the relationship between thunderstorms and the sporadic E Layer over Rome, Astronomische Nachrichten, 334, 968–971, 2013.

Barta, V., Pietrella, M., Scotto, C., Bencze, P., and Sátori, G.: Thunderstorm-related variations in the sporadic E layer around Rome, Acta Geodaetica et Geophysica, 50, 261–270, 2015.

Barta, V., Haldoupis, C., Sátori, G., Buresova, D., Chum, J., Pozoga, M., Berényi, K. A., Bór, J., Popek, M., Kis, Á., et al.: Searching for effects caused by thunderstorms in midlatitude sporadic E layers, Journal of Atmospheric and Solar-Terrestrial Physics, 161, 150–159, 2017.

Davis, C. J. and Johnson, C.: Lightning-induced intensification of the ionospheric sporadic E layer, Nature, 435, 799, 2005.

Davis, C. J. and Lo, K.-H.: An enhancement of the ionospheric sporadic-E layer in response to negative polarity cloud-to-ground lightning, Geophysical research letters, 35, 2008.

Grebowsky, J. M. and Aikin, A. C.: In Situ Measurements of Meteoric Ions, 2002.

Haldoupis, C.: Midlatitude sporadic E. A typical paradigm of atmosphere-ionosphere coupling, Space science reviews, 168, 441–461, 2012.

Haldoupis, C.: Is there a conclusive evidence on lightning-related effects on sporadic E layers?, Journal of Atmospheric and Solar-Terrestrial Physics, 172, 117–121, 2018.

Johnson, C. and Davis, C. J.: The location of lightning affecting the ionospheric sporadic-E layer as evidence for multiple enhancement mechanisms, Geophysical research letters, 33, 2006.

Kopp, E.: On the abundance of metal ions in the lower ionosphere, Journal of Geophysical Research: Space Physics, 102, 9667–9674, 1997.

Mathews, 5 J.: Sporadic E: current views and recent progress, Journal of atmospheric and solar-terrestrial physics, 60, 413–435, 1998

Plane, J.: A time-resolved model of the mesospheric Na layer: constraints on the meteor input function, Atmospheric Chemistry and Physics, 4, 627–638, 2004.

Whitehead, J.: The formation of the sporadic-E layer in the temperate zones, Journal of Atmospheric and Terrestrial Physics, 20, 49–58, 1961.

Whitehead, J.: Recent work on mid-latitude and equatorial sporadic-E, Journal of Atmospheric and Terrestrial Physics, 51, 401–424, 1989.

Yu, B., Xue, X., Lu, G., Ma, M., Dou, X., Qie, X., Ning, B., Hu, L., Wu, J., and Chi, Y.: Evidence for lightning-associated enhancement of the ionospheric sporadic E layer dependent on lightning stroke energy, Journal of Geophysical Research: Space Physics, 120, 9202–9212, 2015.

Yu, B., Xue, X., Lu, G., Kuo, C.-L., Dou, X., Gao, Q., Qie, X., Wu, J., Qiu, S., Chi, Y., et al.: The enhancement of neutral metal Na layer above thunderstorms, Geophysical Research Letters, 44, 9555–9563, 2017.